# Functional Role of NOXA in Hypoxia-Mediated PD-L1 Inhibitor Response in Hepatocellular Carcinoma

**DOI:** 10.3390/ijms26104766

**Published:** 2025-05-16

**Authors:** Mohan Huang, Tian Lan, Xinyue Chen, Rong Chen, Xin Ding, William Chi-Shing Tai, Sze Chuen-Cesar Wong, Lawrence Wing-Chi Chan

**Affiliations:** 1Department of Health Technology and Informatics, The Hong Kong Polytechnic University, Hong Kong SAR, China; mo-han.huang@connect.polyu.hk (M.H.); xinyue24.chen@connect.polyu.hk (X.C.); 2College of Pharmacy, Harbin Medical University, Harbin 150086, China; lantian012345@163.com; 3School of Pharmacy, Guangdong Pharmaceutical University, Guangzhou 510006, China; chenrong9912@163.com (R.C.); dingxin012345@163.com (X.D.); 4Department of Applied Biology and Chemical Technology, The Hong Kong Polytechnic University, Hong Kong SAR, China; william-cs.tai@polyu.edu.hk (W.C.-S.T.); cesar.wong@polyu.edu.hk (S.C.-C.W.)

**Keywords:** hepatocellular carcinoma, hypoxia, immunotherapy resistance, NOXA, hypoxia risk score model

## Abstract

Hypoxia is a crucial characteristic of hepatocellular carcinoma (HCC) and contributes to immune resistance by upregulating PD-L1 and recruiting immunosuppressive cells. However, the molecular mechanisms of hypoxia-induced immunotherapy resistance are still unclear. The hypoxia-related immunotherapy response (IRH) genes were identified and used to develop a hypoxia risk score model to predict patient survival. The model was validated using GSE233802 and EGAD00001008128 datasets. The hypoxia risk score model including NOXA effectively stratified patients based on risk and demonstrated excellent survival predictive ability (*p* = 0.0236). A hypoxia-induced drug-resistant (HepG2-R) cell line was established by co-culturing HepG2 cells with Jurkat T cells under CoCl_2_-induced hypoxia and PD-L1 inhibitor administration. Prolonged exposure to hypoxia (48 h) in HepG2 cells significantly led to the increased hypoxia risk score (*p* < 0.02). The establishment of the HepG2-R cell line showed that prolonged hypoxia reduced cancer cell apoptosis, which implies potential treatment resistance. The effect of NOXA knockdown on the apoptosis of HepG2-R cells under the same co-culture conditions was examined. Under hypoxia and PD-L1 inhibitor treatment, NOXA knockdown increased the survival rate of HepG2-R cells and reduced early and late apoptosis. This indicates that NOXA plays a crucial role in apoptosis regulation and immune response in hypoxic tumors. NOXA knockdown significantly reduces apoptosis in immunotherapy-resistant cells induced by hypoxia. These findings provide important evidence that targeting NOXA may enhance immunotherapy efficacy and help overcome treatment resistance in HCC, highlighting its potential as a therapeutic target.

## 1. Introduction

Hepatocellular carcinoma (HCC) is a leading cause of cancer-related mortality worldwide. Regrettably, most HCC cases are diagnosed at advanced stages. Despite significant advancements in therapeutic strategies, the prognosis for patients with advanced HCC remains poor, with a persistently low five-year survival rate. Notably, over 50% to 60% of these patients require systemic therapies, such as immunotherapy, underscoring the critical role of systemic treatment in managing this malignancy [1,2].

Atezolizumab, a PD-L1 inhibitor, in combination with bevacizumab, a VEGF-targeting antibody, has been approved by the U.S. Food and Drug Administration (FDA) as a first-line treatment for unresectable HCC [3]. In malignant tumor cells, PD-L1 is overexpressed. Atezolizumab binds specifically to PD-L1, disrupting the PD-1/PD-L1 immunosuppressive axis, thereby enhancing the cytotoxicity of antitumor T cells and exerting anticancer effects [4]. The PD-1/PD-L1 signaling pathway is significant in cancer immunotherapy, and inhibitors targeting this pathway have achieved notable therapeutic breakthroughs. However, the current objective response rate of this regimen remains below 30%, with a three-year overall survival rate of less than 50% [1].

Hypoxia is a common feature of solid tumors, arising from an imbalance between oxygen supply and consumption due to continuous tumor cell proliferation [5]. Within the hypoxic tumor microenvironment (TME), hypoxia-inducible factors (HIFs) are activated and serve as pivotal transcription factors regulating cellular responses to low oxygen conditions [6]. Substantial evidence indicates that HIFs play a crucial role in the pathogenesis and pathophysiology of HCC. Under hypoxic conditions, the expression of anti-apoptotic proteins such as Mcl-1 and Bcl-XL increases, while the expression of pro-apoptotic proteins like Bax decreases, facilitating tumor cell survival [7]. NOXA, a pro-apoptotic member of the Bcl-2 protein family, is regulated by HIFs and contributes to the cellular response to hypoxia [8]. In HCC, the interplay between NOXA expression and hypoxic conditions may influence tumor progression and therapeutic outcomes.

In this study, we developed a hypoxia risk score model using bioinformatics and machine learning to assess its association with immunotherapy outcomes. Co-culture experiments were conducted to investigate the effect of NOXA on immune cell activity and the efficacy of PD-L1 inhibitors under hypoxic conditions. By integrating computational modeling with experimental validation, this study aims to elucidate the interplay between hypoxia, NOXA expression, and immune checkpoint blockade efficacy. These findings suggest that modulating NOXA expression may enhance immunotherapy responses, highlighting its potential as a novel biomarker and therapeutic target.

## 2. Results

### 2.1. Identification of IRH and HRH Genes from Public Dataset

The results of this sub-section have been published [9].

As shown in Figure 1a, in the EGAD00001008128 dataset, 84 differentially expressed genes (DEGs) were identified (q-value < 0.05, Up-regulated genes: Fold change > 1.999, Down-regulated genes: Fold change < 1/1.999). As shown in Figure 1b, in the GSE14520 dataset, 400 up-regulated DEGs and 400 down-regulated DEGs were identified (q-value < 0.05, Up-regulated genes: Fold change > 1.139, Down-regulated genes: Fold change < 1/1.1276).

As shown in Figure 1c, in the GSE41666 dataset, 400 up-regulated DEGs and 400 down-regulated DEGs were identified (q-value < 0.05, Up-regulated genes: Fold change > 1.277, Down-regulated genes: Fold change < 1/1.370) and were used for an overlap analysis with the DEGs in the GSE14520 dataset. Among them, 300 up-regulated DEGs were used for an overlap analysis with the DEGs in the EGAD00001008128 dataset (q-value < 0.05, Fold change > 1.3202).

An overlap analysis of the EGAD00001008128 and GSE41666 datasets identified three statistically significant overlapping DEGs (PMAIP1 (NOXA), CD3D, and CD2), which were designated as immunotherapy responses to hypoxia genes (IRHs) (*p* = 0.026, Figure 1d). Similarly, for the GSE14520 and GSE41666 datasets, overlap analysis identified 52 statistically significant overlapping DEGs, which were named HCC-hypoxia overlap genes (HHOs) (*p* = 0.0032, Figure 1e).

### 2.2. Construction of a Hypoxia Risk Scoring Model for HCC

The results of this sub-section have been published [9].

In our previous study, we developed a hypoxia risk scoring model using R packages survival and glmnet. Based on the fragments per kilobase of exon model per million mapped reads (FPKM) data from the Cancer Genome Atlas Liver Hepatocellular Carcinoma (TCGA-LIHC) dataset, univariate COX regression analysis was performed on the HHOs, and 21 genes significantly associated with survival were selected (*p*-value < 0.001, Appendix A). Subsequently, we applied the Least Absolute Shrinkage and Selection Operator (LASSO) regression with cross-validation (the optimal λ value: log(λ) ≈ −4) to further identity and incorporate nine key genes into the hypoxia risk scoring model (Appendix A). Based on previous research, the NOXA gene from the IRHs was also included in the model. The hypoxia risk scoring model, incorporating *PHLDA2*, *DLGAP5*, *N4BP2L1*, *CENPA*, *UPB1*, *CABYR*, *AFM*, *HMMR*, *KIF20A*, and *PMAIP1*, underwent another COX regression analysis using the FPKM dataset of TCGA-LIHC to ensure a significant association with the survival outcomes related to hypoxia (Appendix A). In the formula, the expression of each gene is assigned a coefficient representing its impact on survival. A positive coefficient indicates an increased risk, while a negative coefficient indicates a decreased risk. This formula was used to calculate the hypoxia scores of the GSE233802 and EGAD00001008128 datasets for further validation and analysis of the model.$$\begin{array}{cc} h(t|X) & =h_{0}(t)exp(0.1497\times PHLDA2+0.0757\times DLGAP5\\ & -0.2318\times N4BP2L1+0.0990\times CENPA-0.0584\times UPB1\\ & +0.1509\times CABYR-0.0022\times AFM+0.3139\times HMMR\\ & +0.0587\times KIF20A-0.1203\times PMAIP1) \end{array}$$

### 2.3. Validation of the Hypoxia Risk Scoring Model in Independent Datasets

#### 2.3.1. Hypoxia Risk Scores Reflect Hypoxia Levels in Cell Line Dataset

The model was applied to the GSE233802 dataset, which includes HepG2 cell lines exposed to 0 h, 24 h, and 48 h of hypoxia. Hypoxia risk scores were computed for each time point, showing significant differences (ANOVA, *p*-value < 0.02) in Figure 2a. Scores for the 48 h hypoxia group were higher compared to scores for the 24 h group, while scores at 0 h were higher than at 24 h. These results suggest that early hypoxia reduces risk due to cell proliferation slowdown caused by transient oxygen deprivation. However, prolonged hypoxia significantly increases risk, suggesting a critical transition in tumor adaptation under hypoxic stress.

#### 2.3.2. Hypoxia Risk Stratification Predicts Survival Outcomes in HCC Patients

Patients in the EGAD00001008128 dataset were classified based on their hypoxia risk scores using unsupervised K-Means clustering. This resulted in two groups, as depicted in Figure 2b: a high-risk group comprising 106 patients and a low-risk group of 94 patients. Histogram analysis revealed an approximately normal distribution of the hypoxia risk scores, with cluster centers effectively distinguishing the two subpopulations in Appendix A.

Kaplan–Meier survival analysis demonstrated that patients in the low-hypoxia group exhibited a prolonged PFS compared to those in the high-hypoxia group in Figure 2c. A *p*-value of 0.0236 was obtained from the log-rank test, which meant that the difference in survival outcomes between the two groups was statistically significant. These findings confirm the hypoxia risk stratification model’s effectiveness in differentiating patient survival outcomes.

### 2.4. Establishment of Hypoxia-Induced Resistant HepG2 Cell Line (HepG2-R)

To evaluate the successful establishment of hypoxia-induced drug-resistant cells, we analyzed HIF-1α expression as a key marker of hypoxia. Using Western blot analysis, we measured HIF-1α expression levels under different CoCl_2_ concentrations and exposure durations. As shown in Figure 3a, the levels of HIF-1α protein increased as the concentration of CoCl_2_ rose, and the hypoxic effect was observed within 4 to 48 h of exposure. After co-culturing for 48 h, the HIF-1α expression level in the co-culture group with a concentration of 200 μM CoCl_2_ was still significantly higher than that in the normoxic group. The protein expression levels of HIF-1α in the Western blot images under the conditions of different concentrations of CoCl_2_ and different durations of hypoxia are shown in Appendix A. In subsequent experiments, we used 200 μM CoCl_2_ to induce hypoxia in HepG2 cells.

Flow cytometry analysis revealed that hypoxia-induced HepG2 cells, when co-cultured with varying concentrations of a PD-L1 inhibitor, exhibited a significantly reduced apoptosis rate compared to the normoxic control group. As the duration of exposure to hypoxia lengthened, the anti-apoptotic capacity gradually increased. As shown in Figure 3b, 1-round hypoxia treatment significantly reduced the apoptotic rate compared to the normoxic control group (*p*-value = 0.0005, T-statistic = −4.1431, *t*-test), and 4-round hypoxia treatment resulted in an even greater reduction in apoptosis (*p*-value < 0.0001, T-statistic = −10.7825, *t*-test). The detailed experimental procedure—including the duration and number of hypoxia co-culture cycles applied to HepG2 cells—is illustrated in Method Section 4.6.2, which clarifies the definition of “1 round” and “4 rounds” of hypoxia exposure. The comparison between the 1-round and 4-round hypoxia groups showed a significantly greater reduction in apoptosis in the 4-round hypoxia group (T-statistic = −8.3049, *p*-value < 0.00001, *t*-test). These results indicate that the HepG2 cell line treated with four rounds of hypoxia developed a certain level of anti-apoptotic capability and drug resistance. The HepG2 cell line treated with four rounds of hypoxic co-culture is regarded as the HepG2-R cell line. The gating strategy of flow cytometry is shown in Appendix A.

To optimize the determination of the appropriate concentration of the PD-L1 inhibitor for subsequent experiments, we utilized Nikon’s NIS-Element AR software (Version 6.10.01) to analyze cell proliferation before and after co-culture under microscopic observation. Based on the AI-driven CNN algorithm, the software can identify and quantify the cell fill area on the culture dish before and after co-culture. The system automatically detects fluorescence-positive regions by detecting pixel intensity and morphological features. After co-culture, Jurkat and HepG2 cells adhered to each other, so the software measured the total fluorescent area without separating the two cell types. Since Jurkat cells were seeded at the same number in all wells, the overall increase in fluorescence area mainly reflects the proliferation of HepG2 cells. The analysis results show that the concentration of 0.1 μM PD-L1 inhibitor exhibits the most effective inhibition of cell proliferation. Figure 3c illustrates the change in cell fill area on the culture dish before and after co-culture. The comparison of HepG2 cell fill areas after co-culture across different PD-L1 inhibitor concentrations showed no statistically significant differences. This indicates that the variations in apoptosis rates were not due to differences in baseline cell proliferation among groups but were primarily influenced by the duration of hypoxia exposure.

Under normoxic conditions, a 0.03 μM PD-L1 inhibitor concentration yielded the most pronounced apoptotic effect, as depicted in Figure 3b. This is consistent with the results of a previous study, in which the 0.03 μM PD-1 inhibitor (mAb B1C4) had the most effective immune effect in the co-culture model of Jurkat T cells and HepG2 cells [10]. However, under hypoxic exposure, 0.1 μM PD-L1 inhibitor showed a slightly better performance in apoptosis. This further demonstrates that hypoxia may weaken the efficacy of immunotherapy by disrupting the TME. Since ICIs require an intact TME to achieve their maximum therapeutic effects, the use of different concentrations of PD-L1 inhibitors cannot produce significantly statistically different apoptotic effects. Therefore, 0.1 μM was selected as the optimal concentration for subsequent experiments targeting hypoxia-induced immune resistance.

### 2.5. Efficient Knockdown of NOXA Expression in HepG2-R Cell Line

Western blot and qPCR were used to confirm NOXA knockdown efficiency in HepG2-R cells. This validation step was consistent with the duration of cell co-culture. As shown in Figure 4a, the protein expression levels of NOXA were significantly reduced after transfection with four different siRNAs. siNOXA4 showed the most effective knockdown effect. Compared with the control group, the NOXA protein level was significantly reduced (*p*-value < 0.01). β-Actin was used as an internal control. The qPCR results also confirmed that NOXA mRNA expression was significantly decreased in cells transfected with siNOXA4, further validating the knockdown efficiency, as shown in Figure 4b.

### 2.6. NOXA Knockdown Reduces Apoptosis in HepG2-R Cell Line

To investigate the role of NOXA in apoptosis regulation, NOXA-knockdown (siNOXA) HepG2-R cells were co-cultured with Jurkat T cells under hypoxic conditions. The control group consisted of HepG2-R cells. Apoptosis rates were analyzed using flow cytometry. Figure 5 shows the effect of siNOXA on apoptosis in HepG2-R cells under hypoxic conditions and treatment with 0.1 μM PD-L1 inhibitor. The results indicate that the proportion of live cells increased from 64.10% in the control group to 70.05% in the siNOXA group, suggesting that siNOXA may enhance cell survival in HepG2-R cells under PD-L1 inhibitor treatment. The early apoptosis decreased from 14.77% to 10.50%, while late apoptosis significantly decreased in the siNOXA group, from 16.50% to 10.38%. To further quantify the effect of NOXA knockdown on apoptosis, Cohen’s d values for late and early apoptosis rates are 1.2976 and 1.5109, respectively, both exceeding the 0.8 threshold for large effect size. This indicates that NOXA knockdown has a strong biological effect on reducing apoptosis in HepG2 cells. These results align with previous findings demonstrating that NOXA plays a critical role in regulating apoptosis in HepG2 cells, as reported in the study by Li et al., where NOXA upregulation mitigated tumor growth and promoted apoptosis in an HCC mouse model [10]. These findings suggest that siNOXA in HepG2-R cells may reduce both early and late apoptosis and potentially regulate PD-L1 inhibitor and hypoxia-induced cell death.

## 3. Discussion

Hypoxia is a common feature of the tumor microenvironment. The rapid proliferation of tumors leads to an imbalance between oxygen supply and consumption. Hepatocellular carcinoma (HCC) is one of the most hypoxic tumors, with an intertumoral median oxygen level as low as 0.8%. HCC is a highly vascularized and hypoxic tumor [11,12]. Hypoxia induces molecular and phenotypic changes in cancer cells, primarily through hypoxia-inducible factors (HIFs), especially HIF-1α. Activated HIF-1α can alter gene expression, promoting angiogenesis, metabolic reprogramming, and epithelial–mesenchymal transition, thereby driving tumor progression and metastasis [13]. Hypoxia within tumors can induce the generation of reactive oxygen species (ROS), leading to oxidative stress, increasing DNA damage, and disrupting the DNA repair mechanisms. These changes increase the probability of gene mutations and prompt genetic alterations within tumor cells, resulting in uncontrolled cancer cell proliferation, resistance to apoptosis, and the development of treatment resistance [14]. Many studies have reported that hypoxia increases the expression of the immune checkpoint PD-L1 on the surface of tumor cells, enabling them to enhance immune evasion [14]. These findings highlight the potentially significant impact of hypoxia on HCC progression and treatment response.

In this study, across multiple datasets, we identified some key genes (IRHs and HRGs) related to hypoxia and immunotherapy response in HCC and developed a hypoxia risk score model to predict patient survival [9]. We validated the hypoxia risk score model using the GSE233802 dataset, which includes hypoxic HepG2 cell lines. We found that cells exposed to long-term hypoxia (48 h) exhibited significantly higher hypoxia risk scores compared to those exposed to short-term hypoxia (24 h). This suggests that the internal regulatory mechanisms of tumor cells experience alterations, allowing them to adapt to the hypoxic environment and enhance survival. This adaptation might increase the risk of treatment resistance. The hypoxia risk score model was further validated using the EGAD00001008128 dataset, showing its ability to stratify patients based on hypoxia risk. The progression-free survival (PFS) of patients in the high hypoxia risk group was significantly lower than that of the low-risk group. This result showed the potential of this model in predicting patient survival. Recent studies have also found that the hypoxic tumor microenvironment reduces the efficacy of PD-1/PD-L1 immunotherapy through multiple mechanisms. For example, in glioma cells, HIF-1α can directly bind to the PD-L1 promoter region, enhancing PD-L1 expression [15]. Moreover, hypoxia recruits immunosuppressive cells such as myeloid-derived suppressor cells (MDSCs) and regulatory T cells (Tregs). In hypoxic conditions, these cells selectively upregulate PD-L1 expression by binding HIF-1α to the PD-L1 promoter, which further inhibits T cell activity [16]. Therefore, in our subsequent research, we further explored strategies to overcome hypoxia-induced immunotherapy resistance.

We identified IRHs: PMAIP1 (NOXA), CD3D, and CD2. These genes further emphasize the significant impact of hypoxia on immunotherapy response. Notably, NOXA was included as a key gene in the hypoxia risk score model. NOXA is a pro-apoptotic protein in the Bcl-2 family and plays a crucial role in the regulation of cell apoptosis. NOXA promotes the activation of Bax and Bak by binding to anti-apoptotic proteins such as Mcl-1 and Bcl-xL. This binding leads to mitochondrial outer membrane permeabilization (MOMP), the release of cytochrome c, and the activation of the caspase cascade [17,18,19]. In tumor cells, the expression of NOXA is generally under the regulation of p53. When cells experience stress stimuli such as DNA damage, p53 upregulates NOXA expression, promoting the apoptosis of damaged cells and preventing the accumulation of abnormal cells [20]. In the hypoxic environment, HIF-1α can directly bind to the promoter region of NOXA, leading to its increased expression. This upregulation promotes apoptosis through the mitochondrial pathway [8]. Recent studies have found that higher NOXA levels may enhance the susceptibility of cancer cells to the cytotoxicity mediated by CAR-T cells by promoting the apoptotic pathway. Downregulation or inhibition of NOXA may lead to resistance to this therapy [21]. Furthermore, the NOXA/MCL-1 axis has recently been identified as a critical determinant in cell fate decisions between apoptosis and pyroptosis, highlighting the importance of maintaining a proper balance between pro-apoptotic and anti-apoptotic signals [22,23]. Dysregulation of this balance may contribute to immune resistance and therapeutic failure in HCC. Therefore, understanding the regulatory mechanisms of NOXA in the hypoxic tumor microenvironment is crucial for improving immunotherapy outcomes. 

Although our gene knockdown experiments have shown that reducing the expression of NOXA under hypoxic conditions can decrease the apoptosis of cancer cells, the main purpose of our study is not to imply that inhibiting NOXA is beneficial. Our research findings have revealed the complexity of the role of NOXA: Hypoxia can lead to the upregulation of NOXA expression, and this upregulation is also observed in patients who respond to immunotherapy. This indicates that NOXA may be a key regulatory factor for the response to immunotherapy under hypoxic conditions. The biological effects of modulating NOXA are likely to be related to the tumor microenvironment. For example, they may vary depending on the state of the tumor, the degree of hypoxia, and the immune environment. Future studies involving transcriptomic analysis and mechanistic exploration are needed to further elucidate the downstream signaling pathways involved in the apoptosis of HepG2-R cells mediated by NOXA. Future studies involving transcriptomic analyses and mechanistic exploration will be needed to further explain the downstream signaling pathways involved in NOXA-mediated apoptosis in HepG2-R cells.

We co-cultured activated Jurkat T cells with HepG2 cells under CoCl_2_-induced hypoxic conditions to simulate the hypoxic tumor microenvironment and establish the HepG2-R cell line. Under the same hypoxic conditions and PD-L1 inhibitor treatment, we explored the function of NOXA knockdown in regulating HepG2-R cell apoptosis, revealing its potential as a therapeutic target for improving treatment efficacy. After four rounds of hypoxic co-culture, HepG2 apoptosis was significantly reduced. This indicates that hypoxia might impact immunotherapy effectiveness, as tumor cells can adapt to hypoxic environments and develop treatment resistance. To explore the regulatory role of NOXA in the hypoxic tumor microenvironment and its impact on immune efficacy, we knocked down NOXA in HepG2-R cells. The results showed that under hypoxic conditions with PD-L1 inhibitor treatment, siNOXA increased the proportion of live cells and reduced early and late apoptosis. These findings indicate that NOXA plays a critical role in regulating apoptosis in HepG2-R cells, and its knockdown may enhance cell survival. 

Although knocking down NOXA under hypoxic conditions reduces the apoptosis of cancer cells, this is not in conflict with the association between high NOXA expression and the response to immunotherapy. The high expression of NOXA in patients who respond to immunotherapy indicates that NOXA may play the role of a therapeutic regulator, and its expression may contribute to the apoptosis of cancer cells more effectively through the immune response. The upregulation of NOXA induced by hypoxia most likely represents an inherent attempt to induce the apoptosis of cancer cells. However, in the hypoxic tumor microenvironment, this pro-apoptotic signal may not be sufficient to fully induce the apoptosis of cancer cells. Although hypoxia will induce the expression of NOXA, excessive artificial upregulation may lead complex effects. Therefore, it is important to clarify the role of endogenous NOXA. In the future, modulating the expression of NOXA may help overcome tumor resistance in hypoxic environments and improve immunotherapy outcomes.

## 4. Materials and Methods

### 4.1. Study Design

This study consists of bioinformatics analysis, machine learning, and cell line experiments. Figure 6 shows the overview of the computational hypoxia risk model.

### 4.2. Data Collection and Preprocessing

Several publicly accessible datasets were subjected to bioinformatic analysis, modeling, and cellular experimental validation. These datasets include: EGAD00001008128 from the European Genome-phenome Archive (EGA), downloaded on 10 October 2023, and GSE41666, GSE14520, and GSE233802 from the National Center for Biotechnology Information (NCBI) database, acquired on 26 January 2022, 5 January 2022, and 22 June 2024, respectively. The TCGA-LIHC dataset from The Cancer Genome Atlas (TCGA) database was obtained on 10 January 2024.

#### 4.2.1. Clinical Datasets

The EGAD00001008128 dataset comprises RNA sequencing data from 290 HCC patients’ biopsies. Post-biopsy, these patients received treatment with either atezolizumab or a combination of atezolizumab and bevacizumab. Therapeutic responses were categorized into responders (90 patients) and non-responders (200 patients). The GSE14520 dataset includes microarray data from 214 patients, covering both tumor tissues and adjacent normal tissues, facilitating comparative analyses. The TCGA-LIHC dataset from The Cancer Genome Atlas (TCGA) provides gene expression profiles (FPKM values) and survival data for 367 patients.

#### 4.2.2. Cell Culture Datasets

The GSE41666 dataset includes gene expression profiles of the HepG2 cell line cultures. These cultures were maintained under hypoxic conditions (0% O_2_) and normoxic conditions (21% O_2_) for 24 h. The data were obtained by employing a microarray platform. The GSE233802 dataset contains RNA sequencing data derived from HepG2 cell cultures. These cells were exposed to hypoxia (1% O_2_) for 24 h and 48 h, respectively, with the normoxia set as the control group. In both the GSE41666 and GSE233802 datasets, each experimental condition was repeated three times to ensure biological reproducibility.

#### 4.2.3. Pre-Processing of RNA Sequencing Raw Data

For RNA sequencing data from the EGAD00001008128 and GSE233802 datasets, standard processing pipelines were applied. Quality assessment was performed using FastQC (v0.12.1), while Fastp was used for base trimming and adapter removal. Reads were aligned to the human reference genome (GRCh38.104 from NCBI) using HISAT2, and gene expression was quantified with FeatureCounts, generating the raw expression matrix [24].

To deal with low-quality sequencing results, gene filtering methods from related research were applied. In the analysis of the EGAD00001008128 dataset, for each gene, the median counts per million (CPM) and coefficient of variation (CV) were calculated. Genes below the 25th percentile threshold or expressed in fewer than 75% of samples were excluded [25]. For the GSE233802 dataset, which contains nine samples, only genes with nonzero CPM values in at least six samples were retained.

#### 4.2.4. Normalizing the Data of Gene Expression

The gene expression data, generated using microarray technology from the GSE41666 dataset, had the expression levels of multiple probes per gene averaged. This step resulted in the formation of an expression matrix featuring unique gene symbols. The data went through Variance Stabilizing Normalization (VSN) and were then standardized to a normal distribution, N (0, 1).

For the RNA sequencing datasets EGAD00001008128 and GSE233802, preprocessing was performed using the DESeq2 normalization function (deseq2_norm) in Python (3.8.18). This method applies sample-specific scaling factors to the expression matrices, effectively correcting for sequencing depth and technical variability across samples.

To rescale the EGAD00001008128 expression matrix to a standardized range for further analysis models, the StandardScaler from sklearn in Python was used. This method fits a scaler based on the mean and standard deviation of each gene’s expression. The fitted scaler was then applied to standardize the data to a normal distribution, N (0, 1). For the GSE233802 dataset, each gene’s expression was log-transformed and then standardized to N (0, 1) to validate the subsequent Cox regression model.

### 4.3. Selection of Genomic Features

#### 4.3.1. Differential Expression Analysis

Differential expression analysis (DEA) was performed on the EGAD00001008128, GSE41666, and GSE14520 datasets, using *t*-test q-values and fold change (FC) as evaluation criteria. The statistical significance of each gene was determined by calculating *p*-values. To control the false discovery rate (FDR), q-values were obtained. For the GSE41666 and GSE14520 microarray data, the Storey–Tibshirani method was applied, while for the EGAD00001008128 RNA sequencing data, the Benjamini–Hochberg method was used. Genes were regarded as differentially expressed (DEGs) when they met the criteria of q-value < 0.05 and the thresholds for upregulated or downregulated FC.

#### 4.3.2. Enrichment Analysis

The Venny 2.1 platform (https://bioinfogp.cnb.csic.es/tools/venny/, accessed on 25 May 2024) was used to generate Venn diagrams. These diagrams illustrate the overlap of DEGs between the GSE41666 dataset and the EGAD00001008128 and GSE14520 datasets. The overlapping genes from these datasets were then classified into two separate groups: immunotherapy response to hypoxia (IRH) genes and HCC-hypoxia overlap (HHO) genes. Our previous studies have already reported the HHO genes [26].

### 4.4. Hypoxia Risk Score Model Development

To analyze gene expression (FPKM values) and survival data from HCC patients in the TCGA Liver Cancer (LIHC) database to develop a hypoxia scoring model associated with drug response, we first excluded samples with missing survival time or a survival time of 0 days. This led to a final set of 367 clinical samples being used for training the hypoxia score model. Subsequently, univariate Cox regression analysis (R survival package) was conducted on HHOs, followed by the application of the Least Absolute Shrinkage and Selection Operator (LASSO) algorithm. The R glmnet package was used to perform 10-fold cross-validation to determine the optimal parameter λ and the hypoxia-related genes (previously reported in our studies) [27]. These selected genes were then combined with IRHs to construct the model. The phreg function from Statsmodels library was used to fit the proportional hazards regression model, and regression coefficients for the model genes were calculated from the TCGA dataset.

The model performance was evaluated using the hazard function *λ(t)*, and the baseline hazard function *λ*_0_*(t)* signified the risk level when there were no covariates. The impact on risk was indicated by the covariate coefficient *β_i_*. A positive value meant an increased risk, while a negative value indicated a decreased risk.$$\lambda \left(t\right)=\lambda_{0}(t)\times exp(\beta_{1}\times X_{1}+\beta_{2}\times X_{2}+...\beta_{n}\times X_{n})$$

### 4.5. Model Validation

#### 4.5.1. Calculation of Hypoxia Score

To evaluate tumor risk under varying durations of hypoxia, the model was further verified using the standardized GSE233802 dataset. This dataset contains cell lines that have been exposed to hypoxia conditions for 0, 24, and 48 h. Due to the problems of current clinical techniques in directly measuring intertumoral hypoxia, the model was applied to the standardized EGAD00001008128 dataset to evaluate the risk of hypoxia in HCC patients. The calculated hypoxia risk scores were included in the feature set for further analysis.

#### 4.5.2. K-Means Clustering of Hypoxia Score

To categorize patients in the EGAD00001008128 dataset based on their hypoxia scores, unsupervised K-Means clustering was applied. The patients in the dataset received biopsy and RNA sequencing before receiving immunotherapy. To validate the direct relationship between the hypoxia score and progression-free survival (PFS) rate, patients with an immunotherapy response were excluded from the dataset. The K-Means function from the scikit-learn library was used with clusters = 2 to divide the samples into two clusters. The clustering was performed on the Hypoxia score variable, and the resulting cluster labels were mapped to high- and low-hypoxia groups.

#### 4.5.3. Survival Analysis Using Kaplan–Meier Curves

Kaplan–Meier survival analysis was used to evaluate the survival differences between the high- and low-hypoxia groups in the EGAD00001008128 dataset. The KaplanMeierFitter function from the lifelines package was applied to fit survival curves for both groups. Progression-free survival (PFS) was used as the time-to-event variable. To determine whether the survival distributions of the high- and low-hypoxia groups differed significantly, a log-rank test was performed using the logrank test function from lifelines. A *p*-value < 0.05 was considered statistically significant.

### 4.6. Cell Experiment

#### 4.6.1. Cell Lines and Culture

The hepatocellular carcinoma (HCC) cell line HepG2 (Serial: SCSP-651) and human T lymphocyte cell line (Jurkat T cells; Serial: SCSP-653) were utilized in this study. Both cell lines were procured from the Chinese Academy of Sciences Cell Bank (Beijing, China). HepG2 cells were cultured in Dulbecco’s Modified Eagle Medium (DMEM; Gibco, London, UK; 11965092) supplemented with 10% fetal bovine serum (FBS; Gibco; 10100147C) and 1% penicillin–streptomycin (P/S; Gibco; 15240062). Jurkat T cells were cultured in RPMI-1640 medium (Gibco; C22400500BT) containing 10% FBS and 1% P/S. All cells were incubated at 37 °C in a humidified atmosphere with 5% CO_2_.

#### 4.6.2. Establishment of Co-Culture System

Jurkat T cells were activated with 2 μg/mL phytohemagglutinin (PHA; Thermo, Waltham, MA, USA; R30852701) for 48 h. HepG2 cells were labeled with CellTrace™ reagent (Thermo; C34564) at a 1:1000 dilution (cell density: 1 × 10^6^ cells/mL) and incubated at 37 °C for 30 min. Then, HepG2 cells were seeded into 6-well plates (2 × 10^5^ cells/well) containing DMEM 10% FBS, 1% P/S, 10 ng/mL IFN-γ (Sigma, Saint Louis, MO, USA; IF002) and 200 μM cobalt chloride (CoCl_2_; Sigma; 232696) for 24 h. After that, HepG2 medium was replaced with fresh DMEM containing 10% FBS, 1% P/S, 200 μM CoCl_2_ and PD-L1 inhibitor (PD-L1i; BioXcell, Hyderabad, Telangana; Atezolizumab biosimilar: SIM0009) at concentrations ranging from 0 μM to 1 μM. The activated Jurkat T cells were added to HepG2 cells at an effector-to-target cell (E/T) ratio of 10:1 and co-cultured for 48 h. HepG2 cells were exposed to multiple rounds of hypoxia induction and co-culture. To better determine the appropriate concentration of the PD-L1 inhibitor, the NIS-Element AR software (Version 6.10.01) from Nikon was used to analyze the proliferation rate of cells before and after co-culture. Based on the convolutional neural network (CNN) algorithm of artificial intelligence, this software can identify and calculate the changes in the fill area by cells in the culture dish before and after co-culture. The experimental workflow is shown in Figure 7.

#### 4.6.3. NOXA Gene Knockdown in Hypoxia-Resistant Cells

HepG2 cells resistant to hypoxia (HepG2-R) were seeded at 2 × 10^5^ cells/well in 6-well plates. After 24 h, HepG2-R cells were transfected with four distinct siRNAs targeting NOXA (siNOXA1–4; Jima, Tokyo, Japan; 03625) using Lipofectamine™ 3000 (Thermo; L3000015). The siRNA–Lipofectamine complexes were added to HepG2-R cells in Opti-MEM™ medium (Thermo; 31985070) for 6 h. After that, the medium was replaced with containing DMEM 10% FBS and 1% P/S for 12 h. Untreated HepG2-R cells served as the control group. After transfection, the cell co-culture experiments were conducted. The transfection experimental workflow is shown in Figure 8.

#### 4.6.4. Western Blot Analysis

After co-culture, HepG2 cells and HepG2-R cells were lysed in RIPA buffer (Thermo; 89900) containing protease inhibitors. Protein concentrations were quantified using a Quick BCA Protein Assay Kit (Thermo; 23225). Lysates were denatured in Laemmli buffer (Bio-Rad, Shinagawa City, Tokyo; 1610737) at 98 °C for 10 min, resolved on 12% or 15% SDS-polyacrylamide (SDS-PAGE; Bio-Rad; 1610732) gels. Proteins were transferred to PVDF membranes (Millipore, Burlington, MA, USA; ISEQ00010). The membranes were blocked with 5% skim milk in Tris Buffered Saline with Tween 20 (TBST; Bio-Rad; BUF028) for 1 h and incubated overnight at 4 °C with primary antibodies. The following antibodies were used: anti-HIF-1α (1:4000; Abcam, Waltham, MA, USA; ab179483), anti-NOXA (1:1000; Abcam; ab13654), and anti-β-Actin (1:5000; CST, Boston, MA, USA; 4967S). The membranes were then incubated with horseradish peroxidase (HRP)-conjugated secondary antibodies (1:5000; Thermo; 31462) for 1 h at room temperature. β-Actin was used as a control. Protein bands were visualized using ECL reagent (Bio-Rad; 170-5060) and analyzed with ImageJ software.

#### 4.6.5. RT-qPCR Analysis

Total RNA was extracted using TRIzol reagent (Thermo; 15596018), quantified via NanoDrop (Thermo; ND-2000C), and reverse-transcribed into cDNA (Stemcell, Vancouver, BC, Canada; 79004). The qPCR experiment was performed using a Real-Time fluorescent qPCR instrument (Roche LightCycler 480, Penzberg, Germany) in combination with SYBR Green PCR Master Mix (Vazyme, Nanjing, China, Q411-02/03). Primer sequences are listed in Table 1 (Sangon Biotech, Shanghai, China). Data were normalized to β-Actin and analyzed using the 2−ΔΔCt method.

#### 4.6.6. Apoptosis Assay

The cells (Jurkat T and HepG2 cells) that completed co-culture were collected and resuspended in a binding buffer (1 × 10^6^ cells/mL). The co-culture of HepG2 cells without hypoxia treatment was used as the control group to establish the hypoxia-resistant (HepG2-R) cell line. The co-culture of HepG2-R cells without NOXA knockdown treatment was used as the control group for the transfection experiment. Cell suspensions (100 μL) were stained with 5 μL FITC Annexin V and 5 μL propidium iodide (PI; BD Pharmingen, San Diego, CA, USA; 556547) for 15 min in the dark. After adding 200 μL binding buffer, apoptosis was analyzed using a BD FACSCantoTM II flow cytometer. Data were processed with FlowJo software (v10.8.1).

#### 4.6.7. Statistical Analysis

Software tools, including ImageJ, FlowJo, and Excel, were used to analyze the cell experimental results. The results are presented as the mean ± standard deviation. Statistical analysis was performed using one-way analysis of variance (ANOVA), T-test, and Cohen’s d quantification. A *p*-value < 0.05 was considered statistically significant.

## 5. Conclusions

In conclusion, our research reveals the critical role of hypoxia in regulating the immune response and apoptosis of HCC cells. We established a hypoxia risk score model, developed a hypoxia-resistant HepG2-R cell line, and validated the regulatory role of NOXA in apoptosis within resistant cell lines. This study provides important insights into the immune resistance induced by the hypoxic tumor microenvironment. Regulating NOXA expression has the potential to enhance immunotherapy efficacy and reduce treatment resistance in HCC. Future research should further investigate the molecular mechanisms through which NOXA regulates apoptosis under hypoxic conditions and explore the potential therapeutic effects of combining NOXA inhibition with PD-L1 inhibitors.

## Figures and Tables

**Figure 1 ijms-26-04766-f001:**
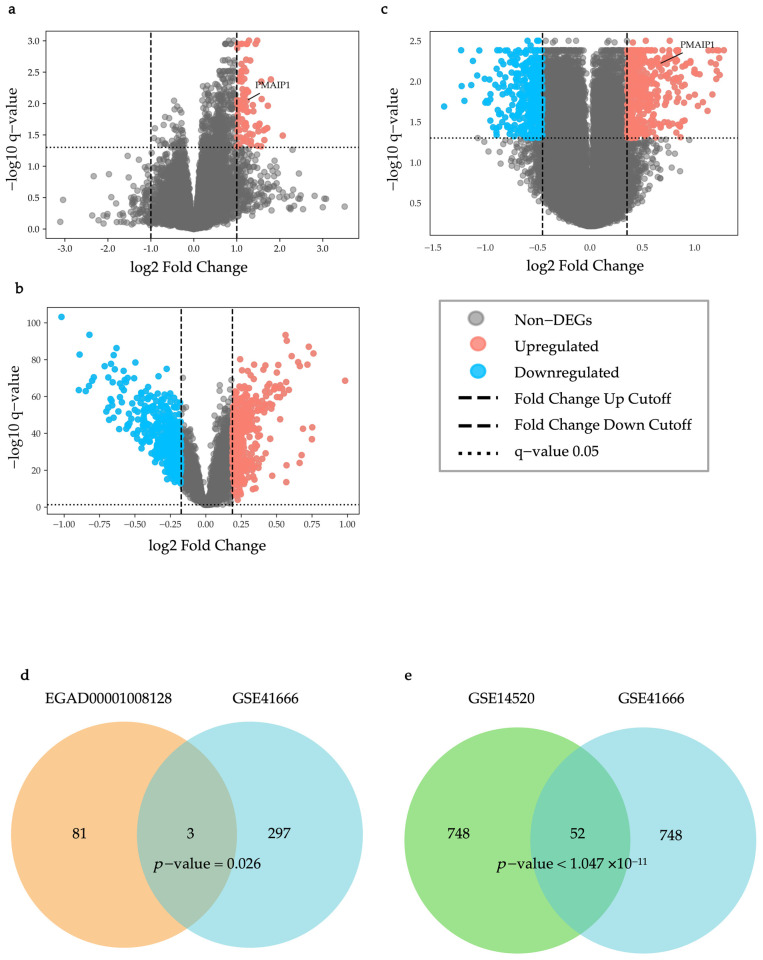
Differential expression analysis. (**a**) A volcano plot for the EGAD00001008128 dataset; (**b**) a volcano plot for the GSE41666 dataset; (**c**) a volcano plot for the GSE14520 dataset; (**d**) IRH genes: the overlapping DEGs between the EGAD00001008128 and GSE41666 datasets; and (**e**) HHO genes: the overlapping DEGs between the GSE41666 and GSE14520 datasets.

**Figure 2 ijms-26-04766-f002:**
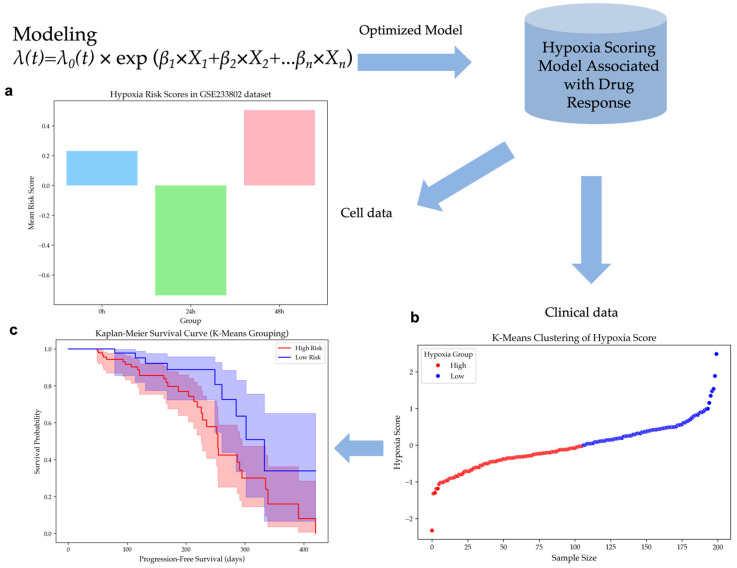
Hypoxia scoring model validation. (**a**) Hypoxia risk scores for HepG2 cells exposed to hypoxia for 0 h, 24 h, and 48 h in the GSE233802 dataset (*p*-value < 0.02). (**b**) K-Means clustering of hypoxia risk scores, dividing patients into high-risk (red) and low-risk (blue) groups in the EGAD00001008128 dataset. (**c**) Kaplan–Meier survival curves between high hypoxia risk and low hypoxia risk patient groups (*p*-value = 0.0236).

**Figure 3 ijms-26-04766-f003:**
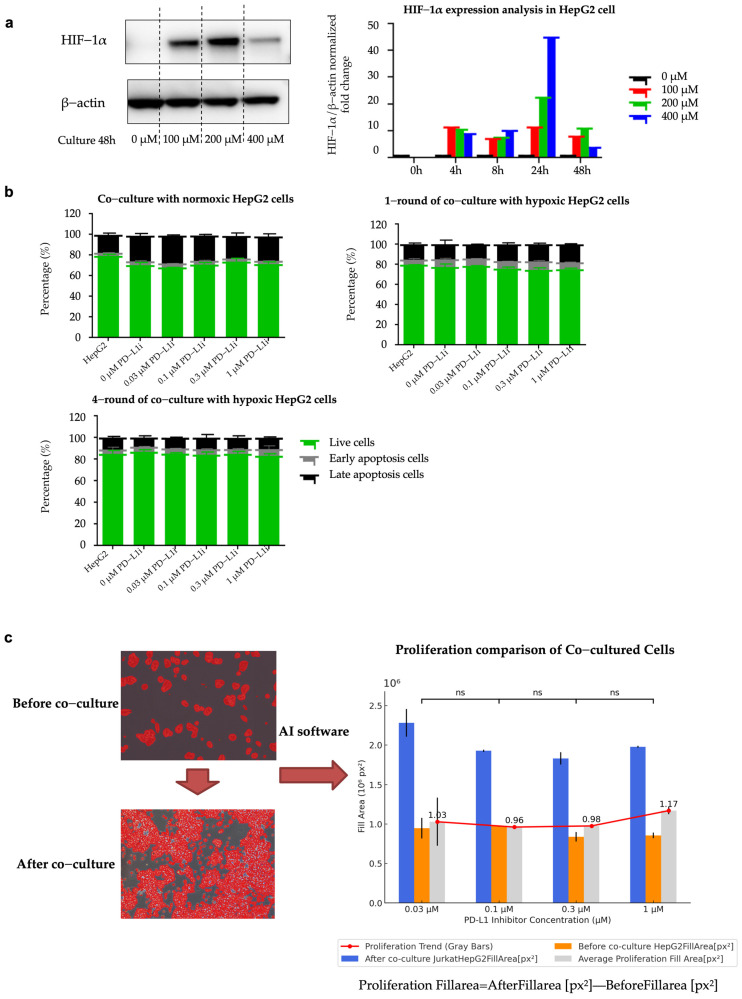
Establishment of hypoxia-induced drug-resistant cells. (**a**) Western blotting was used to detect the effect of CoCl_2_ on the expression of HIF-1α in HepG2 cells (0–400 μM) for 48 h. The grayscale values were analyzed using ImageJ software (version 1.53a). β-Actin was used as the internal control. Protein expression was normalized to the control group. (**b**) Apoptosis analysis of co-culture cell lines. Graphs show the percentage of live cells, early apoptosis cells, and late apoptosis cells in normoxic, hypoxic-treated 1- and 4-round HepG2 cells (normoxic: 1 round hypoxia, *p*-value = 0.0005; normoxic: 4 rounds hypoxia, *p*-value < 0.0001; 1 round hypoxia: 4 rounds hypoxia, *p*-value < 0.00001). (**c**) Proliferation comparison of co-cultured cells under the microscope. Before co-culture, only HepG2 cells were contained. After co-culture, both Jurkat T cells and HepG2 cells were contained. The graph on the right displays the fill area of cells at different PD-L1 inhibitor concentrations.

**Figure 4 ijms-26-04766-f004:**
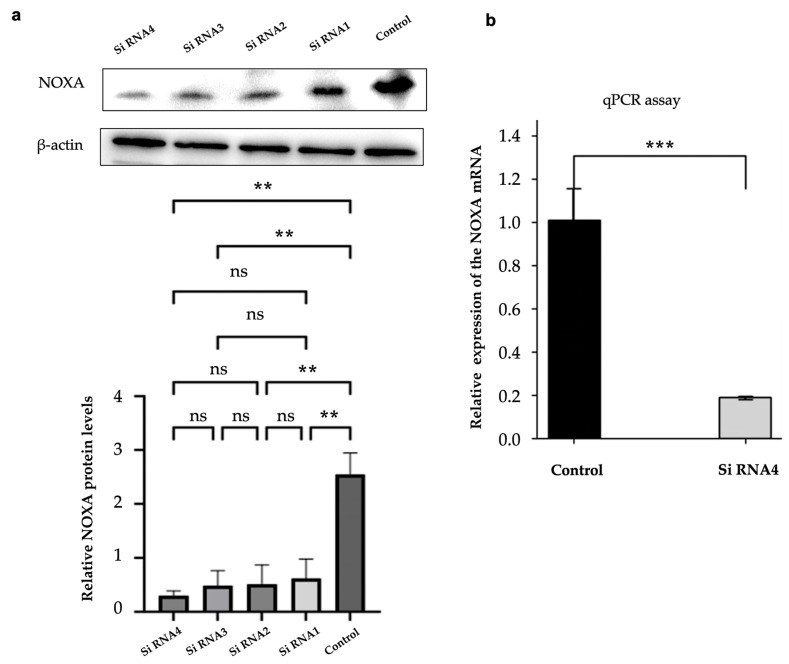
NOXA knockdown efficiency in HepG2-R cells. (**a**) Western blot analysis showing the protein expression levels of NOXA in HepG2-R cells transfected with four different siRNAs. The bar graph below represents the relative NOXA protein levels. (**b**) qPCR analysis of NOXA mRNA expression in HepG2-R cells transfected with siRNA4 and the control group. The data are expressed as mean ± SD, ns *p*-value >0.05, ** *p*-value < 0.01, *** *p*-value < 0.001 vs. control.

**Figure 5 ijms-26-04766-f005:**
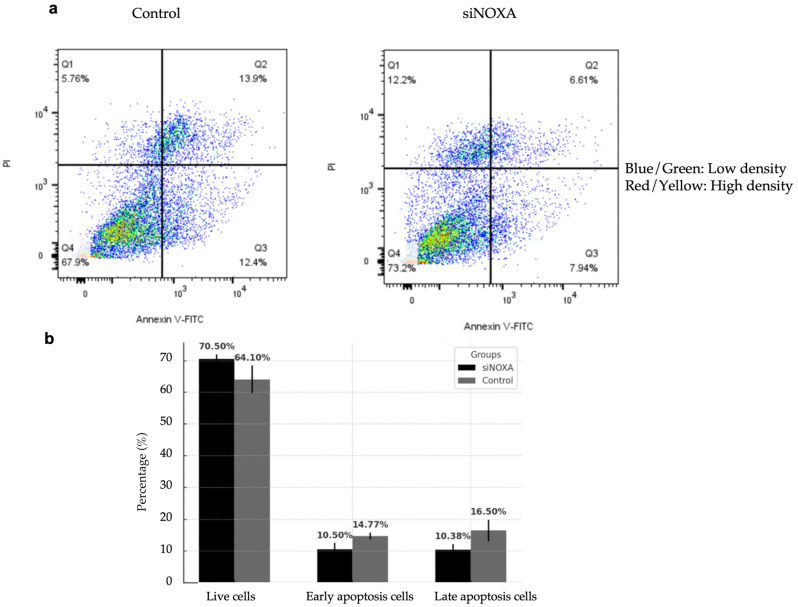
Effect of siNOXA on HepG2-R cell apoptosis. (**a**) Representative flow cytometry analysis showing the distribution of live, early apoptosis, and late apoptosis cells in the control and siNOXA groups. (Blue/Green) indicates low density, while (Red/Yellow) indicates high density. (**b**) Bar chart shows the percentage of live cells, early apoptosis cells, and late apoptosis cells presented as mean ± SD from three independent biological replicates. Cohen’s d value for early apoptosis rate is 1.5109. Cohen’s d value for late apoptosis rate is 1.2976.

**Figure 6 ijms-26-04766-f006:**
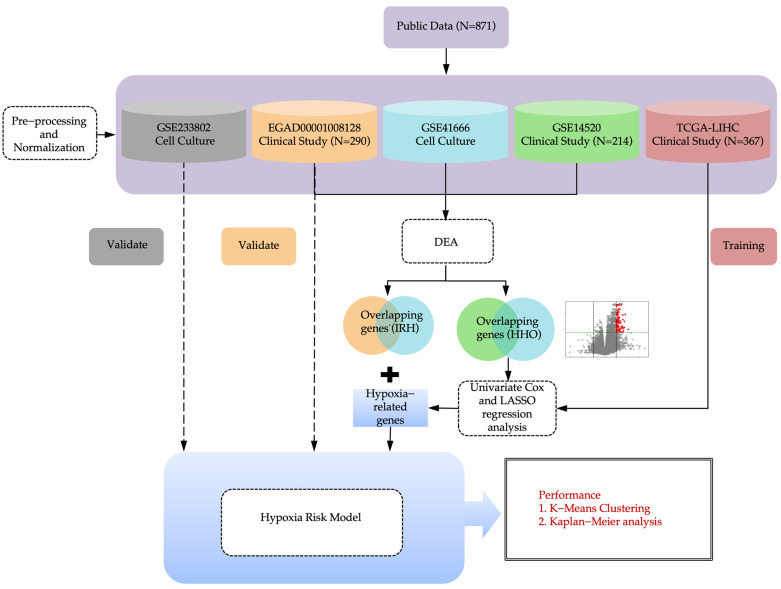
An overview of the computational hypoxia risk model. Dark gray represents the GSE233802 dataset used for cell culture validation; Orange represents the EGAD00001008128 clinical dataset used for clinical validation; Blue represents the GSE41666 cell culture dataset; Green represents the GSE14520 clinical dataset; Light red represents the TCGA-LIHC dataset used for model training through LASSO and Cox regression; Light blue indicates the integrated hypoxia-related genes used for model construction.

**Figure 7 ijms-26-04766-f007:**
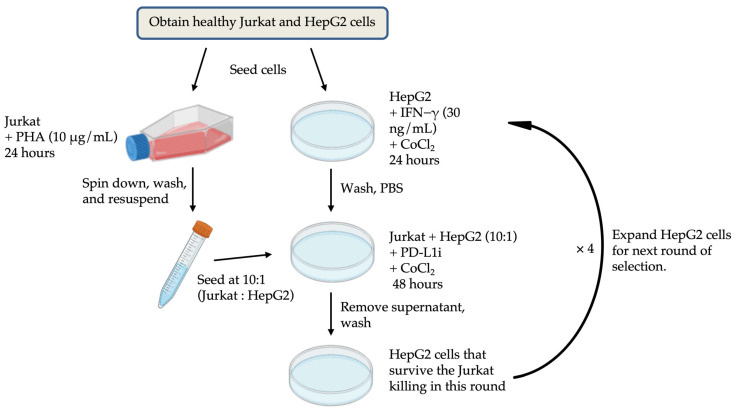
Co-culture flowchart.

**Figure 8 ijms-26-04766-f008:**
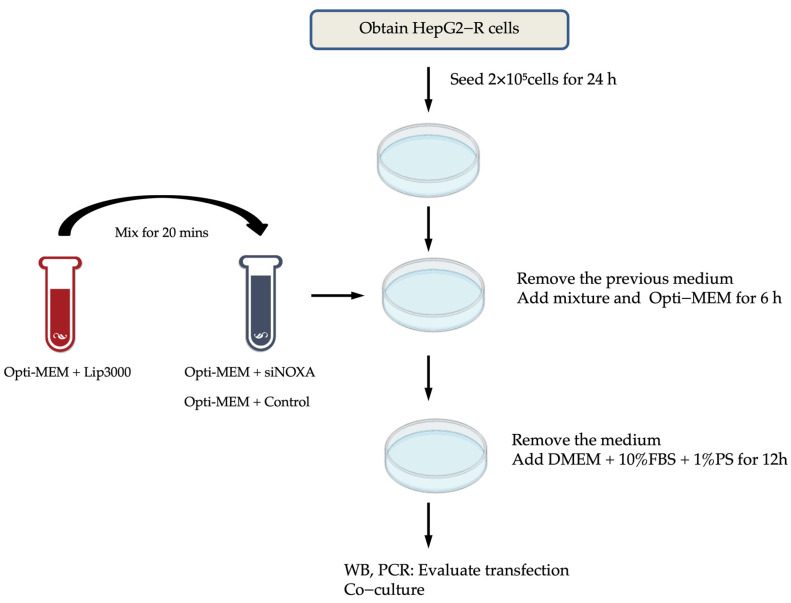
The experimental workflow for NOXA knockdown by transfection in HepG2−R cells.

**Table 1 ijms-26-04766-t001:** The sequences of primers.

Gene	Forward Primer (5-3′)	Reverse Primer (5-3′)
*NOXA*	CAGAGCTGGAAGTCGAGTGTGC	TGCAGTCAGGTTCCTGAGCAGA
*β-Actin*	AGGATTCCTATGTGGGCGAC	ATAGCACAGCCTGGATAGCAA

## Data Availability

https://ega-archive.org/datasets/EGAD00001008128 (accessed on 10 October 2023); https://www.ncbi.nlm.nih.gov/geo/query/acc.cgi?acc=gse14520 (accessed on 5 January 2022); https://www.ncbi.nlm.nih.gov/geo/query/acc.cgi?acc=GSE41666 (accessed on 26 January 2022); https://www.ncbi.nlm.nih.gov/geo/query/acc.cgi?&acc=GSE233802 (accessed on 22 June 2024).

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
