# Peer review of "Functional Role of NOXA in Hypoxia-Mediated PD-L1 Inhibitor Response in Hepatocellular Carcinoma"

_ijms, 2025, doi:10.3390/ijms26104766_

Round 1

Reviewer 1 Report

Comments and Suggestions for Authors

Huang and wo-workers try to address in the manuscript molecular mechanisms related to tumor resistance to present standard of care for the treatment of hepatocellular carcinoma, for which hypoxia and response of the tumor and the micro-environment to hypoxic conditions may play a key role. They initially try to define transcriptomic signatures of hypoxia and immune response to hypoxia in publicly available data sets. They then try to clarify the role of NOXA in the underlying resistance mechanisms, more specifically involvement of NOXA in apoptosis and resistance to treament. 

The study is of high interest for the scientific community, the physicians, the drug makers and eventually the patients. 

The manuscript is well written, even if some parts may beneft from rephrasing/clarifications, the English used is clear and easy to read.

Globally, the approach is revelant, experiment design is sound. 

Nevertheless, the biological effects reported are limited, even if statistically significant according to the analysis reported in the "results" section. Eventually, the conclusions seem to go in the opposite direction of the results reported in figures, and to the previously reported study. 

As such, I cannot approve publication of the manuscript in the present form. 

Best regards

Author Response

  1. I have revised the abstract according to the suggestion.

‘The hypoxia-related immunotherapy response (IRH) genes were identified and used to develop a hypoxia risk score model to predict patient survival. The model was validated using GSE233802 and EGAD00001008128 datasets. The hypoxia risk score model including NOXA effectively stratified patients based on risk and demonstrated excellent survival predictive ability (p = 0.0236). A hypoxia-induced drug-resistant (HepG2-R) cell line was established by co-culturing HepG2 cells with Jurkat T cells under CoCl₂-induced hypoxia and PD-L1 inhibitor administration. Prolonged exposure to hypoxia (48h) in HepG2 cells significantly led to the increased hypoxia risk score (p < 0.02). The establishment of the HepG2-R cell line showed that prolonged hypoxia reduced cancer cell apoptosis, which implies potential treatment resistance. The effect of NOXA knockdown on the apoptosis of HepG2-R cells under the same co-culture conditions was examined. Under hypoxia and PD-L1 inhibitor treatment, NOXA knockdown increased the survival rate of HepG2-R cells and reduced early and late apoptosis.’

  1. I’m sorry, but I didn’t quite understand the suggested revision here.

  1. Please highlight the 2 populations on the graph

Thank you for your suggestion. Figure 2b already clearly displays the two hypoxia risk groups, with red and blue dots representing the high- and low-risk populations, respectively. We now label the high-risk group in red and the low-risk group in blue for better clarity, and this update is included in the revised figure legend. ‘(b) K-Means clustering of hypoxia risk scores, dividing patients into high- (red) and low-risk (blue) groups in the EGAD00001008128 dataset.’

  1. Please refer to fig 7 so one can see what a round or 4 rounds mean

Thank you for pointing this out. To clarify the definition of “1 round” and “4 rounds” of hypoxia treatment, we have added a reference to Figure 7 in the relevant section of the Results. ‘The detailed experimental procedure—including the duration and number of hypoxia co-culture cycles applied to HepG2 cells—is illustrated in Figure 7, which clarifies the definition of “1 round” and “4 rounds” of hypoxia exposure.’

  1. I hardly can see any effect of the PD-L1i on apoptosis which would drive the dose selection.

Thank you for this suggestion. We have revised the Results section to clarify the rationale for PD-L1 inhibitor dose selection. Specifically, we have added a comparison between normoxic and hypoxic conditions, referencing a previous study on PD-1 inhibitor (mAb B1C4) for consistency. The new content also explains how hypoxia affects the tumor immune microenvironment and immunotherapy efficacy, supporting the use of 0.1 μM as the optimal concentration for further experiments. The revised paragraph now reads: Under normoxic conditions, 0.03 μM PD-L1 inhibitor exhibited the best apoptosis effect. This is consistent with the result of a previous study, in which the 0.03 μM PD-1 inhibitor (mAb B1C4) had the most effective immune effect in the co-culture model of Jurkat T cells and HepG2 cells [9]. However, under hypoxic exposure, 0.1 μM PD-L1 inhibitor showed a slightly better performance in apoptosis. This further demonstrates that hypoxia may weaken the efficacy of immunotherapy by disrupting the TME. Since ICIs require an intact TME to achieve their maximum therapeutic effects, the use of different concentrations of PD-L1 inhibitors cannot produce significantly statistically different apoptotic effects. Therefore, 0.1 μM was selected as the optimal concentration for subsequent experiments targeting hypoxia-induced immune resistance.

  1. Please add treatment & concentrations. Is there any control condition?

Thank you for your comment. We have revised the Figure 3a to clearly indicate the treatment conditions and corresponding concentrations of the applied agent. We also have revised Figure 3a legend. ‘(a) Western blotting was used to detect the effect of CoCl₂ on the expression of HIF-1α in HepG2 cells (0 – 400 μM) for 48 h.’

  1. "Before co-culture" is misleading since both cell lines are visibly in the same well.

Thank you for your comment. In the images of "Before co-culture", only HepG2 cells were seeded in the wells. At this stage, Jurkat T cells had not been added yet. After the co-culture started, in the same wells, Jurkat T cells were added and eventually adhered to the HepG2 cells, which made it technically difficult to distinguish or separate them morphologically under the condition of "After co-culture".

  1. What are the staining methods/proteins? I guess HepG2 are red-labeled and Jurkat green?

Thank you for your comment. We would like to clarify that the red and green signals shown in the fluorescence images are derived from the raw fluorescence channel outputs processed by the Nikon NIS-Element AR software. This software employs an AI-driven convolutional neural network (CNN) algorithm, which segments the cell morphology and quantifies the fluorescence-positive regions according to the signal intensity and texture features.

Due to the close adhesion between HepG2 cells and Jurkat T cells after co-culture, the system identifies the total fluorescence-positive region without distinguishing individual cell types. However, the seeding density of Jurkat T cells is the same under all conditions, and they have poor proliferation ability in DMEM. Therefore, the increase in the quantified region mainly reflects the proliferation of HepG2 cells. I have already modified the signal colors in the images, changing them all to red. I have also added a detailed explanation of the software's principle to the article, as follows.

‘The system automatically identifies fluorescence-positive regions by detecting pixel intensity and morphological features. After co-culture, Jurkat and HepG2 cells became attached to each other, so the software measured the total fluorescent area without separating the two cell types. Since Jurkat cells were seeded at the same number in all wells, the overall increase in fluorescence area mainly reflects the proliferation of HepG2 cells.’

  1. What is non-statistically different? HEPG2 fill area before co-culture or average proliferation?

Thank you for this important question. We would like to clarify that “non-statistically different” refers to the comparison of the HepG2 cell fill area after co-culture across different PD-L1 inhibitor concentrations. Specifically, although minor variations in fluorescence area were observed among groups, these differences were not statistically significant. Therefore, the proliferation of HepG2 cells was not notably affected by the PD-L1 inhibitor concentration itself, and the observed differences in apoptosis were mainly attributed to hypoxia exposure rather than baseline proliferation differences. We have revised the text to clearly specify this point.

‘Figure 3c shows the change in cell fill area on the culture dish before and after co-culture. The comparison of HepG2 cell fill areas after co-culture across different PD-L1 inhibitor concentrations showed no statistically significant differences. This indicates that the variations in apoptosis rates were not due to differences in baseline cell proliferation among groups, but were primarily influenced by the duration of hypoxia exposure.’

  1. Is your conclusion that anti–PD-L1 promotes proliferation of hypoxia-resistant HEPG2 cells?

Thank you for this important question. Our conclusion is not that anti–PD-L1 promotes the proliferation of hypoxia-resistant HepG2 cells. Our data suggest that hypoxia exposure, rather than PD-L1 inhibition, played a dominant role in HepG2 cell of apoptosis resistance. We have revised the manuscript to clearly state that the observed effects are attributed mainly to hypoxia-induced changes rather than PD-L1 inhibitor-induced proliferation. This experimental procedure is mainly aimed at finding an appropriate concentration of PD-L1 inhibitor for the subsequent experiments.

‘This indicates that the variations in apoptosis rates were not due to differences in baseline cell proliferation among groups, but were primarily influenced by the duration of hypoxia exposure.’

  1. You could cite 10.7150/jca.70282 to report that your results using hypoxia-resistant HEPG2 are coherent with previously published studies on HepG2.

Thank you for the helpful suggestion. We have carefully reviewed the study (Li et al., 2022, DOI: 10.7150/jca.70282), which investigated the role of NOXA and PUMA genes in promoting apoptosis in hepatocellular carcinoma models. Although our study focuses on NOXA knockdown in hypoxia-resistant HepG2 cells, both our findings and the cited study highlight the critical role of NOXA in regulating apoptosis in HCC. The citation has been added to Section 2.6.

‘These results are consistent with previous findings demonstrating that NOXA plays a critical role in regulating apoptosis in HepG2 cells, as reported in the study by Li et al., where NOXA upregulation mitigated tumor growth and promoted apoptosis in a hepatocellular carcinoma mouse model.’

  1. Why % in a) and b) are different? Is the figure b) an average of two or more replicates/experiments?

Thank you for pointing this out. Figure 3a shows a representative flow cytometry plot from a single experiment. Figure 3b presents the quantitative results averaged from three independent biological replicates. We have revised the figure legend.

‘Figure 5. Effect of siNOXA on HepG2-R cell apoptosis. (a) Representative flow cytometry analysis showing the distribution of live, early apoptosis, and late apoptosis cells in the control and siNOXA groups. (b) Bar chart shows the percentage of live cells, early apoptosis cells, and late apoptosis cells presented as mean ± SD from three independent biological replicates. Cohen's d value for early apoptosis rate is 1.5109. Cohen's d value for late apoptosis rate is 1.2976.’

  1. Repair mechanism are several, so please add an S.

Thank you for the careful reading. We have corrected the wording from “repair mechanism” to “repair mechanisms”.

  1. In Figure 5b, you show that siNOXA decreases apoptosis of HepG2-R, but I guess the objective is eventually to kill tumor cells. So it may be more relevant to increase NOXA expression rather than decreasing it?

Thank you for your suggestion. Under hypoxic conditions, the expression of NOXA has already been upregulated. In this study, we aimed to investigate the regulatory role of NOXA in the process of hypoxia-induced apoptosis. If NOXA is further overexpressed when its expression has already been upregulated due to hypoxia, it will be difficult to distinguish whether the apoptosis is caused solely by the activity of NOXA or by the complex effects resulting from excessive upregulation. Therefore, we chose to knockdown the expression of NOXA to explore its functional role in the hypoxic environment. In the final discussion section, I made the following modifications.

‘Hypoxia will induce the expression of NOXA, and excessive artificial upregulation may lead complex effects. Therefore, it is important to clarify the role of endogenous NOXA. In the future, modulating the expression of NOXA may help overcome tumor resistance in hypoxic environments and improve immunotherapy outcomes.’

  1. I think the culture medium is Opti-MEM.

Thank you for pointing out the mistake. I have corrected the error in Figure 8.

Reviewer 2 Report

Comments and Suggestions for Authors

This study utilized bioinformatics and machine learning to develop a hypoxia risk score model and establish a hypoxia-induced drug-resistant HepG2 cell line for validation. The cell biological validation demonstrates that NOXA involves in hypoxia-mediated resistance to PD-L1 inhibition within this HepG2-R cell line. The following suggestions aim to improve the manuscript.

  1. The subtitles of result section should be revised to more precisely reflect the main findings.

  1. In Figure 1a-c, NOXA should be directly labeled within the volcano plots.

  1. In Figure 3a and Figure S2, sample conditions should be directly labeled above the Western blot images. Additionally, the control group (0 µM for 48 hr) should be included in Figure 3a.

  1. Is NOXA upregulated in HepG2-R cells compared to normal HepG2 cells? The differential expression of NOXA between HepG2-R and HepG2 cells should be presented.

  1. Does NOXA overexpression promote apoptosis in HepG2-R and HepG2 cells?

  1. The mechanism underlying NOXA-mediated modulation of apoptosis in HepG2-R cells should be evaluated through experiments and transcriptomic analysis, or, at least, discussed in the context of hepatocellular carcinoma.

  1. The Methods section requires revision. Catalog numbers for all experimental reagents must be included in this section.

Author Response

Open Review

This study utilized bioinformatics and machine learning to develop a hypoxia risk score model and establish a hypoxia-induced drug-resistant HepG2 cell line for validation. The cell biological validation demonstrates that NOXA involves in hypoxia-mediated resistance to PD-L1 inhibition within this HepG2-R cell line. The following suggestions aim to improve the manuscript.

  1. The subtitles of result section should be revised to more precisely reflect the main findings.

Thank you for the valuable suggestion. We have revised the subtitles in the Results section to more precisely reflect the key findings of each part.

Examples of revised subtitles: ‘Hypoxia Risk Scoring Model’ changed ‘Construction of a Hypoxia Risk Scoring Model for HCC’. ‘Validation of Hypoxic Cell Line Dataset’ changed ‘Hypoxia Risk Scores Reflect Hypoxia Levels in Cell Line Dataset’. ‘Clustering of Hypoxia Risk Scores and Survival Analysis’ changed ‘Hypoxia Risk Stratification Predicts Survival Outcomes in HCC Patients’. ‘Hypoxia-Induced Resistance in HepG2 Cells’ changed ‘Establishment of Hypoxia-Induced Resistant HepG2 Cell Line (HepG2-R)’. ‘NOXA Knockdown Efficiency’ changed ‘Efficient Knockdown of NOXA Expression in HepG2-R Cell Line’. ‘Apoptosis Analysis of NOXA-Knockdown Potentially Resistant Cells’ changed ‘NOXA Knockdown Reduces Apoptosis in HepG2-R Cell Line’.

  1. In Figure 1a-c, NOXA should be directly labeled within the volcano plots.

Thank you for the suggestion. We have revised Figures 1a and 1c by labeling PMAIP1 (NOXA) within the volcano plots. For Figure 1b, the dataset compares tumor tissues with adjacent non-tumor tissues, and PMAIP1 (NOXA) did not show significant differential expression between the two groups. Therefore, no label was added to Figure 1b.

  1. In Figure 3a and Figure S2, sample conditions should be directly labeled above the Western blot images. Additionally, the control group (0 µM for 48 hr) should be included in Figure 3a.

Thank you for the suggestion. We have revised Figure 3a and Figure S2 by labeling the sample treatment conditions. For Figure S2, we have provided detailed annotations of sample IDs, CoCl₂ concentrations, and culture timepoints below the Western blot images in the accompanying table.

  1. Is NOXA upregulated in HepG2-R cells compared to normal HepG2 cells? The differential expression of NOXA between HepG2-R and HepG2 cells should be presented.

Thank you for the suggestion. As presented in the results from the EGAD00001008128 and GSE41666 datasets, NOXA expression was already found to be upregulated in hypoxia-associated high-risk groups and tumor tissues. Therefore, additional comparison between HepG2 and HepG2-R cells was not performed, as the datasets findings already support the upregulation of NOXA under hypoxic conditions.

  1. Does NOXA overexpression promote apoptosis in HepG2-R and HepG2 cells?

Thank you for the suggestion. In this study, we mainly aimed to knockdown NOXA to explore the effects it has on hypoxia-induced immune resistance. We did not conduct experiments involving the overexpression of NOXA. Hypoxia has already upregulated the expression of NOXA. Therefore, experimentally inducing the overexpression of NOXA under hypoxic conditions may trigger complex effects. It will be difficult to distinguish whether apoptosis is caused by exogenous overexpression or by the effects triggered by other factors.

  1. The mechanism underlying NOXA-mediated modulation of apoptosis in HepG2-R cells should be evaluated through experiments and transcriptomic analysis, or, at least, discussed in the context of hepatocellular carcinoma.

Thank you for your valuable suggestion. In this study, we primarily focused on the functional validation of NOXA’s role in apoptosis regulation under hypoxic conditions through siRNA knockdown experiments. Although we did not perform transcriptomic analyses or detailed mechanistic experiments, we have expanded the discussion section to comprehensively review the potential mechanisms by which NOXA modulates apoptosis in hepatocellular carcinoma, based on recent studies.

‘This binding leads to mitochondrial outer membrane permeabilization (MOMP), the release of cytochrome c, and the activation of the caspase cascade [16-18]. Furthermore, the NOXA/MCL-1 axis has recently been identified as a critical determinant in cell fate decisions between apoptosis and pyroptosis, highlighting the importance of maintaining a proper balance between pro-apoptotic and anti-apoptotic signals [21,22]. Dysregulation of this balance may contribute to immune resistance and therapeutic failure in HCC. Future studies involving transcriptomic analyses and mechanistic exploration will be needed to further explain the downstream signaling pathways involved in NOXA-mediated apoptosis in HepG2-R cells.’

  1. The Methods section requires revision. Catalog numbers for all experimental reagents must be included in this section.

Thank you for your suggestion. We have carefully revised the methods section to include the catalog numbers for all commercially available experimental reagents, including antibodies, chemical inhibitors, and assay kits.

‘HepG2 cells were cultured in Dulbecco’s Modified Eagle Medium (DMEM; Gibco; 11965092) supplemented with 10% fetal bovine serum (FBS; Gibco; 10100147C) and 1% penicillin-streptomycin (P/S; Gibco; 15240062).’

‘Then, HepG2 cells were seeded into 6-well plates (2 × 10⁵ cells/well) containing DMEM 10% FBS, 1% P/S, 10 ng/mL IFN-γ (Sigma; IF002) and 200 μM cobalt chloride (CoCl₂; Sigma; 232696) for 24 h. After that, HepG2 medium was replaced with fresh DMEM containing 10% FBS, 1% P/S, 200 μM CoCl₂ and PD-L1 inhibitor (PD-L1i; BioXcell; Atezolizumab bio-similar: SIM0009) at concentrations ranging from 0 μM to 1 μM.’

Round 2

Reviewer 2 Report

Comments and Suggestions for Authors

The revised manuscript adequately addresses most previous concerns. It’s clarity, conciseness, and overall quality have been substantially enhanced. The revised manuscript is now suitable for publication in IJMS.

Author Response

Comments and Suggestions for Authors The revised manuscript adequately addresses most previous concerns. It’s clarity, conciseness, and overall quality have been substantially enhanced. The revised manuscript is now suitable for publication in IJMS. Response to Comments and Suggestions: We sincerely thank the reviewer for your positive feedback. We highly appreciate the constructive comments and valuable suggestions provided throughout the review process. We are pleased that the revised manuscript has addressed the previous concerns and is now considered suitable for publication.